# From Natural Xanthones to Synthetic C-1 Aminated 3,4-Dioxygenated Xanthones as Optimized Antifouling Agents

**DOI:** 10.3390/md19110638

**Published:** 2021-11-13

**Authors:** Diana I. S. P. Resende, Joana R. Almeida, Sandra Pereira, Alexandre Campos, Agostinho Lemos, Jeffrey E. Plowman, Ancy Thomas, Stefan Clerens, Vitor Vasconcelos, Madalena Pinto, Marta Correia-da-Silva, Emília Sousa

**Affiliations:** 1CIIMAR—Interdisciplinary Centre of Marine and Environmental Research, University of Porto, Terminal de Cruzeiros do Porto de Leixões, Avenida General, Norton de Matos S/N, 4450-208 Matosinhos, Portugal; dresende@ff.up.pt (D.I.S.P.R.); joana.reis.almeida@gmail.com (J.R.A.); sandra.c.pereira28@gmail.com (S.P.); amoclclix@gmail.com (A.C.); vmvascon@fc.up.pt (V.V.); madalenakijjoa@gmail.com (M.P.); esousa@ff.up.pt (E.S.); 2Laboratory of Organic and Pharmaceutical Chemistry, Department of Chemical Sciences, Faculty of Pharmacy, University of Porto, Rua Jorge Viterbo Ferreira, 228, 4050-313 Porto, Portugal; up201002662@ff.up.pt; 3Department of Biology, Faculty of Sciences, University of Porto, Rua do Campo Alegre, 4069-007 Porto, Portugal; 4AgResearch Ltd., 1365 Springs Rd, Lincoln 7674, New Zealand; Jeff.Plowman@agresearch.co.nz (J.E.P.); Ancy.Thomas@agresearch.co.nz (A.T.); Stefan.Clerens@agresearch.co.nz (S.C.); 5Biomolecular Interaction Centre, University of Canterbury, Christchurch 8041, New Zealand; 6Riddet Institute, Massey University, Palmerston North 4474, New Zealand

**Keywords:** xanthones, anti-settlement, antifouling, molecular targets, eco-friendly alternatives

## Abstract

Biofouling, which occurs when certain marine species attach and accumulate in artificial submerged structures, represents a serious economic and environmental issue worldwide. The discovery of new non-toxic and eco-friendly antifouling systems to control or prevent biofouling is, therefore, a practical and urgent need. In this work, the antifouling activity of a series of 24 xanthones, with chemical similarities to natural products, was exploited. Nine (**1**, **2**, **4**, **6**, **8**, **16**, **19**, **21**, and **23**) of the tested xanthones presented highly significant anti-settlement responses at 50 μM against the settlement of mussel *Mytilus galloprovincialis* larvae and low toxicity to this macrofouling species. Xanthones **21** and **23** emerged as the most effective larval settlement inhibitors (EC_50_ = 7.28 and 3.57 µM, respectively). Additionally, xanthone **23** exhibited a therapeutic ratio (LC_50_/EC_50_) > 15, as required by the US Navy program attesting its suitability as natural antifouling agents. From the nine tested xanthones, none of the compounds were found to significantly inhibit the growth of the marine biofilm-forming bacterial strains tested. Xanthones **4**, **6**, **8**, **16**, **19**, **21**, and **23** were found to be non-toxic to the marine non-target species *Artemia salina* (<10% mortality at 50 μM). Insights on the antifouling mode of action of the hit xanthones **21** and **23** suggest that these two compounds affected similar molecular targets and cellular processes in mussel larvae, including that related to mussel adhesion capacity. This work exposes for the first time the relevance of C-1 aminated xanthones with a 3,4-dioxygenated pattern of substitution as new non-toxic products to prevent marine biofouling.

## 1. Introduction

Marine biofouling, a process in which submerged surfaces are colonized by micro-and macro-organisms, has been an increasingly serious problem to several industrial sectors around the world, causing vast economic losses and increasing security risks [1]. Particularly for the shipping industry, the attachment of marine organisms on ship hulls causes decreased speed performance, leading to increasing fuel consumption and consequently pollution [2]. Therefore, biofouling harmful effects are also extendable to the environment and, consequently, to human health. These are affected either by the spread of aquatic invasive species or by enlarged fuel consumption and therefore, by extension, the CO_2_ emissions.

The need for improved strategies to reduce the impact of biofouling is urgent toward the development of alternative non-toxic and environmentally friendly antifouling agents. A considerable amount of research has been conducted on the use of natural products with biological functionality as alternative antifoulants [3,4,5,6,7,8,9]. However, a group of secondary metabolites, both from marine and terrestrial sources, which is still largely unexplored for antifouling purposes are xanthone derivatives. Four marine xanthones were described for the first time in 2013 as potential antifouling compounds. Yicathins B and C (Figure 1), isolated from *Aspergillus wentii* pt-1 [10], exhibited antimicrobial activity (yicathin B was active against *Escherichia coli*, and yicathin C could inhibit *E. coli*, *Staphylococcus aureus*, and *Colletotrichum lagenarium*) [10], and sterigmatocystin and methoxysterigmatocystin, isolated from the fungus *Aspergillus sp*. [11], revealed to be potent inhibitors of the larval settlement of *Balanus amphitrite* (EC_50_ < 0.125 µg.mL^−1^) [11]. Regarding toxicity, while yicathins B and C exhibited weak brine shrimp (*Artemia salina*) toxicity [10], it was observed that sterigmatocystin and methoxysterigmatocystin induced paralyzing effects on the barnacle larvae [11]. Two years later, a new marine xanthone was described with antifouling activity [12]; penicilixanthone (Figure 1) was isolated from the marine-derived fungus *Aspergillus terreus* SCSGAF0162 [12] and revealed inhibitory activity against the larval settlement of *B. amphitrite* (EC_50_ = 17.1 µg.mL^−1^).

In an attempt to further explore the potential of xanthone derivatives for antifouling purposes, we explored for the first time the antifouling activity of a series of nineteen synthetic analogs [13]. 3,4-Dihydroxyxanthone, also a naturally occurring xanthone, exhibited in vivo activity toward the settlement of *Mytilus galloprovincialis* larvae, low toxicity to this macrofouling species, and no ecotoxicity toward *A. salina* (<10% of mortality) [13].

3,4-Dihydroxyxanthone fulfills several properties of a perfect antifouling agent; the higher EC_50_, compared to Econea^®^, challenged us to search for new derivatives with improved potency. In this direction in this work, we proceeded with efforts to explore the structure–antifouling activity relationship (SAR) of xanthones and achieve optimized xanthone-derived antifoulants. Our findings revealed that the presence of hydroxyl groups at positions 3 and 4 was not crucial for activity since derivatives with methoxy groups at these positions were also active [14]. In this work, a library of 24 synthetic xanthones with three substitution patterns—1,3,4-trisubstituted xanthones (**1**, **6**, **8**–**10**, **12**–**24**), 1,3,4,6-tetrasubstituted xanthones (**2**, **4**, **5**, **7**, and **11**) and 1,2,3,4,6-pentasubstituted xanthone (**3**)—was studied. Additionally, the xanthones selected for this work display a wide diversity of substituents at C-1—three xanthones with a methyl group (**1**, **2**, **3**), four with halogenated alkyl groups (**4**–**7**), one with an hydroxymethyl group (**8**), three with an aldehyde (**9**–**11**), and 14 with diverse aminated alkyl groups (**12**–**24**) (Figure 2).

Anti-settlement bioassays against the adhesive larvae of the macrofouling mussel (*M. galloproviancialis*) and antibacterial assays against five biofilm-forming marine bacteria (*Cobetia marina*, *Vibrio harveyi*, *Pseudoalteromonas atlantica*, *Halomonas aquamarina* and *Roseobacter litoralis*) were performed to assess xanthones **1**–**24** antifouling potential. Compounds exhibiting promising antifouling effects were evaluated for their marine ecotoxicity using *Artemia salina*, and the molecular effects after exposure of *M. galloprovinciallis* plantigrade larvae were assessed by differential analyses of the proteome to infer antifouling molecular targets.

## 2. Results and Discussion

### 2.1. Synthesis

Xanthones are aromatic oxygenated heterocyclic molecules, with a dibenzo-γ-pyrone scaffold known as 9*H*-xanthen-9-one [15]. Four distinct approaches were developed and optimized over the last years toward the synthesis of this scaffold [16]. In this work, xanthones **1** and **2** were previously synthesized in our laboratory, in the scope of other projects, via the benzophenone method from 1,2,3-trimethoxy-5-methylbenzene and the appropriate methoxybenzoyl chloride (Figure 1) [17,18]. Xanthones **3****–11** were also previously obtained in the scope of other projects through straightforward transformations such as chlorination [17], bromination and solvolytic displacement of the bromine atoms [18]. Regarding aminated xanthones **12**–**24**, were resynthesized for this work via a reductive amination of 3,4-dimethoxy-9-oxo-9*H*-xanthene-1-carbaldehyde (**10**) and the appropriate amines [19].

### 2.2. Structure-Anti-Settlement Activity Relationship of Xanthone Derivatives

To evaluate the antifouling activity of a series of 24 xanthone derivatives, the anti-settlement of *Mytilus galloprovincialis* larvae was selected as the first screening assay. This blue mussel is a particularly important fouling species with high recruitment rates and the ability to outcompete other species. From the series of xanthone derivatives tested, **1**, **2**, **4**, **6**, **8**, **16**, **19**, **21**, and **23** presented highly significant anti-settlement responses at 50 μM (*p* < 0.05) against the settlement of *M. galloprovincialis* larvae when compared to the negative control (Figure 3).

At first glance, comparing 1,3,4,6-substituted xanthones **2**, **7**, and **11** with their respective 1,3,4-substituted xanthones **1**, **6**, and **10**, the introduction of an additional methoxy group at position 6 seems not to affect the anti-settlement activity (**1** vs. **2**, both presenting similar effects) or reduce the anti-settlement activity (xanthones **7** and **11** with an additional methoxy group at position 6 revealed higher percentage values of settlement of mussel larvae than xanthones **6** and **10**). Comparing 1,2,3,4,6-substituted pattern (xanthone **3**) with 1,3,4,6-substituted pattern (xanthone **2**), the introduction of a chlorine atom at position 2 in xanthone **3** nullified the significant anti-settlement effect of xanthone **2**. Comparison of 3,4-dimethoxy xanthones **10**, **21**, and **19**, with 3-methoxy and 4-hydroxy xanthones **9**, **15**, and **12**, respectively, led to the conclusion that the presence of methoxy groups at 3 and 4 positions induced stronger inhibition of the larvae settlement in xanthone **10** (vs. **9**), xanthone **21** (vs. **15**), and xanthone **19** (vs. **12**). Overall, xanthones containing a hydroxy group at position 4 and a methoxy group at position 3 (**9**, **12**–**15**) were not significantly active (*p* > 0.05) against the settlement of *M. galloprovincialis* larvae. Neither CHBr_2_ (**7**) or CHO (**11**) moiety at C-1 produced an enhancement of the anti-settlement activity (65–45% settlement at 50 µM) when compared to the parent xanthone **2**.

Considering a wide diversity of substituents at C-1, the presence of a chloromethyl group in xanthone **4** was associated with an enhancement of the activity (only 20% of settlement) compared to the results obtained with the bromomethyl group in xanthone **5** (45% settlement) and hydroxymethyl group in xanthone **8** (35% settlement). Considering the aminated alkyl groups in **16**–**24**, higher anti-settlement activities were observed for derivatives containing cyclic amine moieties such as morpholine (**16**, 0% settlement), piperazine (**19**, 10% settlement), piperidine (**23**, 20% settlement), or aniline (**21**, 20% settlement). In a previous work, the presence of piperidine in a thioxanthone was also associated with a high anti-settlement activity [13].

From this series of compounds, xanthones **1**, **2**, **4**, **6**, **8**, **16**, **19**, **21**, and **23** were considered promising compounds against the settlement of mussel larvae and were selected from this primary screening for concentration-response analysis (Table 1). All the compounds exhibited EC_50_ values lower than the US navy recommendation (EC_50_ < 25 µg.mL^−1^) [20]. Xanthone **23** emerged as the most effective larval settlement inhibitor (EC_50_ = 3.57 µM; 1.26 µg.mL^−1^), followed by xanthones **21** (EC_50_ = 7.28 µM; 3.03 µg.mL^−1^) and **1** (EC_50_ = 11.81 µM; 3.19 µg.mL^−1^). Additionally, **23** exhibited therapeutic ratios (LC_50_/EC_50_) of higher than 15, which is in accordance with the standard requirement for efficacy level of natural antifouling agents as established by the US Navy program [20].

Overall, SAR analysis confirmed the importance of a 3,4-dimethoxy substitution in the xanthone core with an additional cyclic amine moiety at C-1 (preferably a piperidine, like in **21** and **23**), rendering a hit optimization from the original 3,4-dihydroxyxanthone [13].

### 2.3. Antibacterial Activity

Despite the major importance of macrofouling species in what concerns biofouling prevention, the substrate colonization in marine biofouling occurs primarily by microorganisms that form biofilms, which can influence the subsequent biofouling community [22]. Therefore, microfouling inhibition must be considered when developing efficient antifouling compounds [23].

Therefore, the antibacterial activity of the nine most promising antifouling xanthones **1**, **2**, **4**, **6**, **8**, **16**, **19**, **21**, and **23** was evaluated against five strains of marine biofilm-forming bacteria: *Cobetia Marina*, *Vibrio harveyi*, *Halomonas aquamarina*, *Pseudoalteromonas atlantica* and *Roseobacter litoralis* (Figure 4). None of the tested compounds were able to significantly inhibit (> 30% inhibition) the growth of *C. marina*, *V. harveyi, P. atlantica*, *and H. aquamarina*. Only xanthones **6** and **16** presented a growth inhibition higher than 30% inhibition of the strain *R. litoralis* in the screening bioassays at 12.5 µM (Figure 4); however, this bioactivity was not confirmed for the two compounds in the concentration–response studies (Appendix A) at concentrations between 2 and 32 µM. According to these results, these compounds should not be considered as antibacterial agents, at least against these biofilm-forming strains.

### 2.4. Marine Ecotoxicity

The marine ecotoxicity of the most promising antifouling compounds (**1**, **2**, **4**, **6**, **8**, **16**, **19**, **21**, and **23**) was evaluated using the nauplii of *Artemia salina* at 50 and 25 µM (Figure 5).

In the presence of xanthones **4**, **6**, **8**, **16**, **19**, **21**, and **23**, non-toxic effects (<10% mortality) were observed, even at 50 µM, to this non-target species. Compounds **1** and **2** showed significant toxicity to *A. salina* even with the lowest concentration tested (25 μM) (Figure 5). Considering this preliminary ecotoxicity evaluation, xanthones **4**, **6**, **8**, **16**, **19**, **21**, and **23** may have lower toxicity to the environment than some of the current biocides in use, as is the case of ECONEA^®^ that induced 100% of mortality at the same concentration range [24].

In contrast to our previous results in which a thioxanthone with a piperidine moiety revealed 100% mortality to this non-target marine organism at 50 µM [13], xanthone **23**, with similar potency against *Mytilus* larvae (EC_50_ ca 4 µM), exhibited less than 10% of mortality for *A. salina* at the same concentration.

### 2.5. Insights on the Antifouling Mechanism of Action of the Hit Xanthones ***21*** and ***23***

#### Antifouling Targets by Proteomics

The analysis of the proteome of plantigrade larvae, using the shotgun methodology, led to the identification of 215 proteins. Following this result, a mixed model ANOVA analysis (*p* < 0.05) detected alterations in expression of 156 and 159 proteins, respectively in the groups exposed to compounds **21** and **23**. The profile of expression of this group of proteins is reported in Figure 6A, using a heatmap representation. In the heatmap, lines identify the proteins and their relative expression in the three sample groups (control, groups exposed to compounds **21** and **23**). Proteins are grouped in clusters and the distances between clusters reflect the differences in the profile of expression of these proteins. The columns identify the three experimental groups compared. In Figure 6A, the group exposed to compound **21** is clustered further away from the control, suggesting increased quantitative differences in protein expression in this group. Moreover, color differences from the heatmap indicate that most proteins were downregulated in the groups exposed to compounds **21** and **23** (further protein quantitative and functional information are presented in Appendix A). The Venn diagram (Figure 6B) reveals that 140 proteins were affected by both compounds. In turn, 16 proteins were affected only by compound **21** and 19 proteins by compound **23**. Overall, the Venn diagram suggests that the two compounds have similar molecular targets and affect similar molecular processes.

To improve the perception of the molecular processes affected by the compounds, differentially expressed proteins were classified using gene ontology (GO) terminology and grouped according to their functions (Table 2). The functional classes were searched using the web tool g: Profiler and validated applying a hypergeometric distribution significance test (Fisher’s exact test, *p* value < 0.01) and multiple testing corrections (counts and sizes, g: SCS) (detailed results of GO term search are reported in Appendix A). Figure 7 depicts all functional classes identified.

In Figure 7, classes of related functions are distributed close to each other, forming networks or nodes. Four nodes are recognized. Node 1 group: the molecular functions related with cellular component assembly and biogenesis. Proteins included in this node are Histone H1-delta, histone H4, protein SET (constituents of the chromatin and nucleosome), 40S ribosomal protein, clathrin (constituent of small transport vesicles), plastin-3 (component of actin filament) and villin-1, a protein involved in the reorganization of microvillar actin filaments and actin filament capping, polymerization, and severing. Villin-1 was affected exclusively by compound **21**. Node 2 group: several functional classes related to cilium movement. Associated to these functional classes are several cilia components (tektin proteins and radial spoke head proteins 9 and 4). Radial spoke head proteins were affected only by compound **21**. In mussels, cilia are responsible for driving water flow through filtration for respiration and feeding along the ventral groove toward the mouth. The effects of compounds on cilia integrity might interfere with the health status of mussels [25], contributing to an energetic investment in vital functions instead of production of byssal threads for settlement. Proteins related with structural constituents of the cytoskeleton, including ciliary microtubules units (tubulin alpha and beta chains, and radial spoke head proteins) were previously found to be downregulated in *M. galloproviancialis* larvae exposed to the natural compounds portoamides [23] and associated to an anti-settlement response.

Node 3 group: the functional classes related to translation and metabolism of organonitrogen compounds. Among the proteins involved in these functions are several 60S and 40S ribosomal proteins, the eukaryotic initiation factor 4A, elongation factor 1-gamma and elongation factor 2. Finally, Node 4 group: several functional classes related to the metabolism of small molecules. Proteins with these functions include ATPase synthase complex proteins (ATP synthase subunit gamma, subunit beta and subunit alfa) and enolase-phosphatase E1 that participates in L-methionine biosynthesis. Associated to this node are also proteins of the tricarboxylic acid cycle (citrate synthase, malate dehydrogenase, and glutamate dehydrogenase), glycolysis and gluconeogenesis (fructose-bisphosphate aldolase, enolase), and pentose phosphate pathway (6-phosphogluconate dehydrogenase decarboxylating). Other proteins in this node include nucleoside diphosphate kinase A, involved in the synthesis of nucleoside triphosphates other than ATP, adenosylhomocysteinase, which is an enzyme with an important role in regulating cellular metabolism, via regulation of the intracellular concentration of adenosylhomocysteine and control of methylations and a putative alcohol dehydrogenase class-3. This enzyme is involved in many cellular processes including response to redox state, ethanol and fatty acid oxidation, catabolism of formaldehyde.

Other proteins affected by these compounds play different roles in the adhesion mechanism of mussels. Their expression is shown in Figure 8. Several myosin isoforms from pedal retractor muscle were downregulated by compounds **21** and **23**. Pedal retractor along with protractor muscles, control the movement of mussel foot, the main organ responsible for the synthesis and release of byssal threads. The downregulation of these proteins is therefore speculated to affect foot functions including animal adhesion ability. Along with myosin, the alteration in other structural proteins (actin and α-actinin) could contribute to impair pedal retractor muscle functions. Conversely, the synthesis of new byssal threads, a high energy consuming process, can be compromised with the downregulation of cytochrome c oxidase, a key enzyme from the respiratory chain. Procollagen-Proline Dioxygenase or Prolyl 4-hydroxylase is an enzyme that catalyzes the post-translational formation of 4-hydroxyproline that is found in abundance in byssal proteins (Mfps and pre-cols) in mussels [26]. This modification is important to confer the adhesive properties to byssal proteins and for the assembling of byssal threads. Heat shock-like proteins have been found associated to byssus structures [27]. These proteins are speculated to carry chaperone functions in byssal proteins, contributing to correct assembling of this group of proteins during the process of synthesis of byssal threads.

## 3. Materials and Methods

### 3.1. Chemicals

Xanthones **1**–**11** were previously synthesized in our laboratory in the scope of other projects and used without further purification after purity assessment [17,18,19,28]: 3,4-dimethoxy-1-methyl-9*H*-xanthen-9-one (**1**), 3,4,6-trimethoxy-1-methyl-9*H*-xanthen-9-one (**2**) [17,18], 2-chloro-3,4,6-trimethoxy-1-methyl-9*H*-xanthen-9-one (**3**), 1-(chloromethyl)-3,4,6-trimethoxy-9*H*-xanthen-9-one (**4**) [17], 1-(bromomethyl)-3,4,6-trimethoxy-9*H*-xanthen-9-one (**5**), 1-(dibromomethyl)-3,4-dimethoxy-9*H*-xanthen-9-one (**6**), 1-(dibromomethyl)-3,4,6-trimethoxy-9*H*-xanthen-9-one (**7**) [18] 1-(hydroxymethyl)-3,4-dimethoxy-9*H*-xanthen-9-one (**8**) [19], 4-hydroxy-3-methoxy-9-oxo-9*H*-xanthene-1-carbaldehyde (**9**), 3,4-dimethoxy-9-oxo-9*H*-xanthene-1-carbaldehyde (**10**) [19], 3,4,6-trimethoxy-9-oxo-9*H*-xanthene-1-carbaldehyde (**11**) [18]. Xanthone derivatives **12**–**24** were resynthesized, and their structures were in accordance with previously described procedures [17,18,19,28]: 4-hydroxy-1-((4-(2-hydroxyethyl)piperazin-1-yl)methyl)-3-methoxy-9*H*-xanthen-9-one (**12**) (83%), 1-((4-acetylpiperazin-1-yl)methyl)-4-hydroxy-3-methoxy-9*H*-xanthen-9-one (**13**) (79%), 4-hydroxy-3-methoxy-1-((2-(pyridin-4-ylmethyl)hydrazineyl)methyl)-9*H*-xanthen-9-one (**14**) (88%) [18,28], 1-((5-amino-3,4-dihydroisoquinolin-2(1*H*)-yl)methyl)-4-hydroxy-3-methoxy-9*H*-xanthen-9-one (**15**) (87%) [18], 3,4-dimethoxy-1-(((2-morpholinoethyl)amino)methyl)-9*H*-xanthen-9-one (**16**) (59%), 1-(((3-(dimethylamino)propyl)(methyl)amino)methyl)-3,4-dimethoxy-9*H*-xanthen-9-one (**17**) (63%), 1-(((4-chlorobenzyl)amino)methyl)-3,4-dimethoxy-9*H*-xanthen-9-one (**18**) (61%), 1-((4-(2-hydroxyethyl)piperazin-1-yl)methyl)-3,4-dimethoxy-9*H*-xanthen-9-one (**19**) (69%), 4-((3,4-dimethoxy-9-oxo-9*H*-xanthen-1-yl)methyl)piperazin-2-one (**20**) (37%), 1-((5-amino-3,4-dihydroisoquinolin-2(1*H*)-yl)methyl)-3,4-dimethoxy-9*H*-xanthen-9-one (**21**) (35%), 1-(((2-(diethylamino)ethyl)amino)methyl)-3,4-dimethoxy-9*H*-xanthen-9-one (**22**) (51%), 3,4-dimethoxy-1-(piperidin-1-ylmethyl)-9*H*-xanthen-9-one (**23**) (62%), and 1-(((1-(4-chlorophenyl)ethyl)amino)methyl)-3,4-dimethoxy-9*H*-xanthen-9-one (**24**) (63%) [19]. The purity of compounds **1**–**24** was assessed by HPLC-DAD and was higher than 95%.

### 3.2. Antifouling Screening

#### 3.2.1. *Mytilus galloprovincialis* Larvae Anti-Settlement Bioassays

The 24 synthesized xanthones **1**–**24** were first screened for anti-settlement activity toward *M. galloprovincialis* plantigrade larvae. For the bioassays, *M. galloprovincialis* juveniles were collected in intertidal mussel beds during low tides in Memória beach, Matosinhos, Portugal (41°13′59″ N; 8°43′28″ W). Mussel plantigrade larvae (0.5–2 mm) were sorted in a binocular magnifier (Olympus SZX2-ILLT, Tokyo, Japan) and those presenting foot exploring behavior for settlement were selected to the screening bioassays at 50 µM performed in 24-well microplates for 15 h, in the dark at 18 ± 1 °C, according to Almeida et al. [29]. Four well replicates per condition and five competent larvae per well were included. Negative and positive controls were included, namely DMSO (0.1%) and CuSO_4_ (5 µM) as an efficient antifouling agent. Stock solutions in DMSO (50 mM) were used to prepare test media (50 µM) in filtered seawater (0.1% DMSO). After the exposure period, the anti-settlement bioactivity was determined by the observation of efficiently attached plantigrade larvae considering the existence/absence of produced byssal threads for each test condition. Compounds producing significant settlement inhibition at 50 µM when compared to the negative control were selected for antifouling effectiveness (semi-maximum response concentration that inhibited 50% larval settlement (EC_50_)) and toxicity [median lethal dose (LC_50_) and therapeutic index) bioassays.

#### 3.2.2. Biofilm-Forming Marine Bacteria Growth Bioassays

Five strains of marine biofilm-forming bacteria from the Spanish Type Culture Collection (CECT): *Cobetia marina* CECT 4278, *Vibrio harveyi* CECT 525, *Halomonas aquamarina* CECT 5000, *Pseudoalteromonas atlantica* CECT 570, and *Roseobacter litoralis* CECT 5395 were selected for antibacterial screening of the most promising anti-settlement compounds Phylogenetically different bacterial strains were selected to have a broader representation of different susceptibilities regarding antifouling promising compounds, and at the same time, were represented in biofouling films characterized in the Portuguese coast [30]. The experimental procedure was performed as previously reported by Almeida et al. [21,29]. Briefly, bacteria were inoculated and incubated for 24 h at 26 °C in marine broth (Difco) at an initial density of 0.1 (OD600) in 96 well flat-bottom microtiter plates and exposed to the test compounds at 15 μM. Bacterial growth inhibition in the presence of the compounds was determined in quadruplicate at 600 nm using a microplate reader (Biotek Synergy HT, Winooski, VT, USA). Negative and positive controls used were a solution of marine broth with 0.1% DMSO, and a stabilized solution of marine broth with penicillin-streptomycin-neomycin, respectively. Compounds producing bacterial growth inhibition higher than 30% were considered as hits and were selected for further dose-response studies (2–32 µM), for estimation of EC_50_ if applicable.

### 3.3. Antifouling Molecular Targets Assessment in M. galloprovincialis Larvae by Differential Proteome Analysis

#### 3.3.1. Sample Preparation

Samples for proteomic analysis were prepared according to Antunes et al. [23]. Briefly, ten mussel larvae from the anti-settlement bioassays were used per replicate. In total four replicates per condition were analyzed. The biological material was incubated in lysis buffer with 2% (*w*/*v*) SDS, 100 mM Tris-HCl, 0.1 M DTT and protease inhibitors (Roche, 11697498001, Basel, Switzerland), pH 7.6 for 1 h with mixing (450 rpm) and further heated at 95 °C, 3 min for protein denaturation. All samples were clarified at 16,000× *g* for 20 min and total protein was estimated based on the absorbance at 280 nm (1 Abs = 1 mg.mL^−1^ protein) with Nanodrop (Thermo Scientific, Waltham, MA, USA) before storage (−80 °C). Proteins were subsequently digested following filter aided sample preparation (FASP) [31] with modifications. Protein samples (100 µg) were alkylated with iodoacetamide, incubated with trypsin (Roche, 03708985001) using 1:50 enzyme to protein ratio (*w*/*w*) for 16 h at 37 °C, in 30 kDa nominal molecular weight limit (NMWL) centrifugal filter units (MRCF0R030, Millipore, Billerica, MA, USA). Subsequently, excess detergent was removed with Thermo Scientific spin columns (reference number 87777). Cleaned samples were acidified with formic acid, dried in a Centrivap centrifugal concentrator (Labconco, Kansas City, MO, USA) and finally resuspended in 50 µL of 0.1% formic acid and 2% acetonitrile.

#### 3.3.2. LC-MS/MS Analysis

LC-MS/MS was performed on a nanoflow Ultimate 3000 UPLC (Dionex) coupled to an Impact HD mass spectrometer equipped with a CaptiveSpray source (Bruker Daltonik, Bremen, Germany. For each sample, 1 µL of the sample was loaded on a C18 PepMap100 nano-Trap column (300 µm, ID × 5 mm, 5 micron 100 Å) at a flow rate of 3000 nL.min^−1^. The trap column was then switched in line with the ProntoSIL C18AQ analytical column (100 µm, ID × 150 mm, 3 micron 200 Å). The reversed phase elution gradient was from % to % to 45% B over 60 min, total 85 min at a flow rate of 1000 nL.min^−1^. Solvent A was LCMS-grade water with 0.1% formic acid; solvent B was LC-MS-grade ACN with 0.1% FA. The mass spectrometer was set up in positive ionization MS mode with a mass range of *m*/*z* 130–2200. All samples (including biological repeats) were analyzed in duplicate. To identify the peptides of interest, a pool from each treatment group was made by combining 5 µL of every sample digest and duplicates were run on the Impact HD mass spectrometer. More specifically, the pooled samples were measured in data-dependent MS/MS mode, where the acquisition speed was 2 Hz in MS and 1–20 Hz in MS/MS mode depending on precursor intensity. Ten precursors were selected in the *m*/*z* range of 350–1200, with preference for doubly or triply charged peptides. The analysis was performed in positive ionization mode with a dynamic exclusion of 60 s.

#### 3.3.3. Protein Identification and Quantification

Following LC-MS/MS analysis, the quadrupole time-of-flight (QTOF) data were searched using the Peaks Studio 8.5 search algorithm (Bioinformatics Solutions, Waterloo, ON, Canada) against an in-house Mytilus sequence database and the uniprot-mollusca-200718–290-109 sequence database. The following parameters were used: 1—mass tolerance of 0.15 Da and a fragment mass tolerance of 0.15 Da, 2—trypsin as the digestive enzyme, and 3—up to 2 missed cleavages. Carbamidomethyl (C) was specified as a fixed modification, and oxidation (M) and deamidation (NQ) were chosen as variable modifications. A false discovery rate (FDR) of 1% was used for peptide identification in Peaks. In addition, the Peptide Hit Threshold (−10log_) was set to 30, de novo only 15% of ALC, and only proteins with a minimum of 1 unique peptide identification were included.

#### 3.3.4. Protein Functional Analysis

Protein functional classification (Gene Ontology, GO) was carried out using the g:Profiler web server version e102_eg49_p15_7a9b4d6 with database updated on 15 December 2020 [31]. Proteins were firstly blasted by running the local BLASTp function from Blast2Go program version 5 (basic) against the Crassostrea gigas sequences (39,917 sequences, downloaded from UniProt KnowledgeBase by December 2020), setting a cut-off e value of 1e-3. The homologous C. gigas sequences were then grouped into functional classes (GO terms), employing the g:GOSt function in g:Profiler. Functional classification was carried out using C. gigas genome and the Ensemble database as data source. A hypergeometric test (Fisher’s exact test) was employed to detect statistically significantly enriched biological processes and pathways in the input protein lists. Results were validated by setting a *p* value < 0.01 and performing multiple testing corrections using set counts and sizes (g:SCS) correction method [32].

### 3.4. Ecotoxicity Assessment

Ecotoxicity of the most promising xanthones to non-target species was assessed using the brine shrimp (*Artemia salina*) lethality test standard protocol [21,33]. Newly hatched nauplii I larvae were used for the exposure to the most promising antifouling compounds at 50 and 25 µM. The bioassay was performed in 96-well microplates with eight replicates per condition and 15–20 nauplii per well, including a positive (K_2_Cr_2_O_7_ at 13.6 µM) and negative control (0.1% DMSO). The percentage of nauplii mortality for each condition was determined at the end of the exposure period (48 h).

### 3.5. Data Analysis

Data from the anti-settlement screening were analyzed using a one-way analysis of variance (ANOVA), followed by a Dunnett test against the negative control (*p* < 0.05). Semi-maximum response concentration that inhibited 50% larval settlement (EC50) and the median lethal dose (LC50) for each bioactive compound were assessed using Probit regression analysis (Log10) with 95% lower and upper confidence limits [95% LCL;UCL]. Therapeutic ratios (LC_50_/EC_50_) were used to evaluate the effectiveness versus toxicity of bioactive compounds. In proteomic analysis, the mixed model ANOVA was utilized to report protein abundance differences between treatments at a confidence level of 95%. The mixed model ANOVA was performed on log10(x + 1)-transformed values from 4 biological replicates, employing the software IBM SPSS Statistics (version 21, SPSS Inc., Chicago, IL, USA).

## 4. Conclusions

Preventive strategies toward biofouling are commonly biocide based, which is harmful to the aquatic environment due to its toxic effects. New understanding of biofouling mechanisms created a great opportunity for the development of new antifouling solutions. Environmentally compatible alternatives, such as bioinspired antifoulants with high specificity and high efficiency, have emerged as a more sustainable way to produce new agents to be further applied in the antifouling industry. Due to the size of surfaces where the antifoulant is applied, the bioactive compound must be available in large quantities to be incorporated in the desired formulation. The use of synthetic derivatives instead of natural products can be considered as an advantage, since these can be produced in large scale and in a short period of time. Following structural patterns exhibited by natural antifoulant xanthones, in this work, C-1 aminated xanthones with a 3,4-dioxygenated pattern of substitution were disclosed as new nontoxic products to prevent marine biofouling. Two hit compounds (xanthones **21** and **23**) emerged with antifouling potential regarding anti-settlement effectiveness and low toxicity to mussel larvae at environmentally relevant concentrations. The substitution pattern of xanthones **21** and **23** induced higher anti-settlement activity against Mytilus larvae than our previously studied and most promising and non-toxic synthetic xanthone, 3,4-dihydroxyxanthone, without losing the low toxicity to the environment, highlighting the benefits of the rationale followed in this work. Proteomic studies regarding antifouling targets suggest that xanthones **21** and **23** have similar molecular targets and affect similar molecular processes, including several myosins from pedal retractor muscle. The downregulated proteins from xanthones from this work are different from the downregulated proteins from previous 3,4-dihydroxyxanthone (collagen proteins (PreCols) and proximal thread proteins (TMPs)).

Further studies on the biodegradability of these xanthones in various environmental conditions and the toxicity of the transformation products to guarantee low impact to the environment are currently being performed. Additionally, to confirm the safety of these compounds toward other non-target marine organisms, further toxicity studies should be performed complementarily to *Artemia salina* ecotoxicity. Nevertheless, this study demonstrates the high potential of the studied xanthones as new antifouling agents.

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
