# Peer review of "From Natural Xanthones to Synthetic C-1 Aminated 3,4-Dioxygenated Xanthones as Optimized Antifouling Agents"

_marinedrugs, 2021, doi:10.3390/md19110638_

Round 1

Reviewer 1 Report

This manuscript, submitted to marine drugs by Marta Correia-da-Silva’s group for the special issue, describes antifouling activities, antibacterial activities, ecotoxicities and antifouling mechanism of action with 24 synthetic xanthones. The synthetic xanthones are previously reported by the same group. In the antifouling activity test with mussel, synthetic compounds have shown the possibility of antifouling toward two phases of the mussel, plantigrade and larval. Other biological evaluations are important for the development of xanthone-based antifouling material. The insight of the mechanism of action with xanthones are valuable results for further development of antifouling compounds. With these results in consideration, this referee believe that this manuscript should be accepted after improvement of the following points.

  1. please add general introduction concerning wide background of this research because it is too short and simplified.
  2. Too many ‘Error! Reference source not found.’ please improve these errors.
  3. please use ‘hydroxy’ and ‘methoxy’ instead of ‘hydroxyl’ and ‘methoxyl’ because of mistakes.
  4. line 86: antifouling (AF)
  5. please provide carbon numbers of xanthone in compound 1.
  6. In ref 15, this referee could not find any description about US navy program about antifouling standards. Please check again.
  7. Figure 3: why no reference (A)? And, in the caption, xanthones 1-25 should be 1-24.
  8. Table 1: all values about LC50/EC50 should be >8.47 for compound 1 because all LC50 values all are more than 100 µM.

Author Response

This manuscript, submitted to marine drugs by Marta Correia-da-Silva’s group for the special issue, describes antifouling activities, antibacterial activities, ecotoxicities and antifouling mechanism of action with 24 synthetic xanthones. The synthetic xanthones are previously reported by the same group. In the antifouling activity test with mussel, synthetic compounds have shown the possibility of antifouling toward two phases of the mussel, plantigrade and larval. Other biological evaluations are important for the development of xanthone-based antifouling material. The insight of the mechanism of action with xanthones are valuable results for further development of antifouling compounds. With these results in consideration, this referee believe that this manuscript should be accepted after improvement of the following points.

Point 1: please add general introduction concerning wide background of this research because it is too short and simplified.

Response 1: A more detailed background was included in the Introduction section as suggested.

Point 2: Too many ‘Error! Reference source not found.’ please improve these errors.

Response 2: We were made aware by the Editor that these errors are editor's layout mistakes. They were already fixed.

Point 3: Please use ‘hydroxy’ and ‘methoxy’ instead of ‘hydroxyl’ and ‘methoxyl’ because of mistakes.

Response 3: ‘hydroxyl’ and ‘methoxyl’ were corrected to ‘hydroxy’ and ‘methoxy’ throughout the entire manuscript.

Point 4: Line 86: antifouling (AF)

Response 4: antifouling (AF)” was corrected at line 86.

Point 5: Please provide carbon numbers of xanthone in compound 1.

Response 5: Numbers were added to compound 1 carbons.

Point 6: In ref 15, this referee could not find any description about US navy program about antifouling standards. Please check again.

Response 6: We appreciate the notice. Reference 15 (now 20) was replaced by the correct one: “Qian, P. Y., Xu, Y. & Fusetani, N. Natural products as antifouling compounds: recent progress and future perspectives. Biofouling 2010, 26, 223, doi:10.1080/08927010903470815.”.

Point 7: Figure 3: why no reference (A)? And, in the caption, xanthones 1-25 should be 1-24.

Response 7: We appreciate the notice. We have now improved the Figure 3, 4 and 5 changing (B) and (C) to (C-) and (C+). Furthermore, “xanthones 1-25” was changed to “xanthones 1-24” throughout the entire manuscript. We appreciate the notice.

Point 8: Table 1: all values about LC50/EC50 should be >8.47 for compound 1 because all LC50 values all are more than 100 µM.

Response 8: We appreciate the notice. The symbol “>” was added to all values at LC50/EC50 column.

Reviewer 2 Report

Review comments for marinedrugs-1440504

The manuscript " From natural xanthones to synthetic C-1 aminated 3,4-dioxy-genated xanthones as optimized antifouling agents" researched the antifouling activities of 24 modified known xanthones. The authors investigated their premilary structure-activity relationships, anti-settlement activity, anti-settlement activity and marine ecotoxicity. In addition, two compounds affected similar molecular targets and cellular processes in mussel larvae.

I suggest that this manuscript could be received in Marine Drugs after major revision.

  1. Why did the authors choose CuSO4 as positive control in anti-settlement activity? Metal ions are no longer recommended because of their possible adverse effects on marine organisms. Moreover, the compounds synthesized in this paper belong to organic compounds, so it is more appropriate to choose the same type of commercially available antifouling agent Seanine211. It is suggested to supplement the data of Seanine211.
  2. Is the antifouling activity of carbonyl groups on the skeleton necessary? It is suggested that the carbonyl group in parent xanthone 2 be reduced to hydroxyl group to investigate the complete structure-activity relationships. What is the mass and yield of each compound? Although it has been reported in the literature, a synthetic roadmap should also be added. Moreover, it is very strange for the authors to synthesize all the reported and known derivatives of the antifouling activities? The authors should explain in the manuscript.
  3. According to the insights on the antifouling mode of action of the hit xanthones 21 and 23,it is recommended that the authors supplement molecular docking data for compounds and molecular targets associated with mussel adhesion, rather than merely conjectural.
  4. "Synthesis" is not appropriate as a keyword. There is hardly any description of synthesis in the article. Please replace or supplement the part of synthetic chemistry.
  5. In the anti-settlement activity test, why choose such a high concentration as 50 μM for screening? It should decrease the concentration for screening, sometimes high concentration will inhibit the situation.
  6. Please add the MIC values of 1, 2, 4, 6, 8, 16, 19, 21, and

Obviously, different compounds exhibit different activity on different indicators, so if possible, all compounds should be tested for antimicrobial activity.

  1. The “Materials and Methods” part is lack of “Antibacterial Activity”.
  2. There are some other problems that the authors should check carefully.

Line 31/150: What does 15 refer to in the passage?  Please explain clearly. If it is different from the xanthones 15, it is suggested to use another expression method.

Line 32: When the abbreviation first appears in the manuscript, please give the full name, for example, AF.

Line 87/91: Please use the same expression, numbers or words. And check the number of xanthones.

Line 95: Why you choose these five kinds of bacteria for antibacterial assays? Please supplement the evidence.

Line 113: The results in Figure 3 are difficult to be compared between groups. An intuitive statistical method is recommended to compare differences between groups.

Line 130: The difference of data between group C and group 16 cannot be shown in Figure 3. It is suggested to do it again and indicate the repetition times under the figure. The error lines of some groups are too high. Are the results reliable?

Line 172: It is recommended to mark significant differences of 6 and 16 in Figure 1. Why not set 12.5μL concentration in the concentration-response of antibacterial activity?  Supplements are suggested to validate the results.

Line 222: The names of proteins are not fully displayed, so the heat map of Figure 6A is suggested to be placed separately. Important proteins are recommended to be compared on separate heat maps.

Full text: It is recommended to mention figure or graph is analyzed in the result statement.

Author Response

The manuscript " From natural xanthones to synthetic C-1 aminated 3,4-dioxy-genated xanthones as optimized antifouling agents" researched the antifouling activities of 24 modified known xanthones. The authors investigated their premilary structure-activity relationships, anti-settlement activity, anti-settlement activity and marine ecotoxicity. In addition, two compounds affected similar molecular targets and cellular processes in mussel larvae.

I suggest that this manuscript could be received in Marine Drugs after major revision.

Point 1: Why did the authors choose CuSO4 as positive control in anti-settlement activity? Metal ions are no longer recommended because of their possible adverse effects on marine organisms. Moreover, the compounds synthesized in this paper belong to organic compounds, so it is more appropriate to choose the same type of commercially available antifouling agent Seanine211. It is suggested to supplement the data of Seanine211.

Response 1: We are aware that CuSO4 is not applicable as AF agent in real protective systems, however it is a potent AF agent frequently used as test validation (positive control) in laboratory trials. However, we understand the reviewer concern and we have added the EC50 of an organic biocide (Econea), previously published by us in the same bioassay conditions (Neves et al., 2020), in table 1 for comparative purposes.

Point 2: Is the antifouling activity of carbonyl groups on the skeleton necessary? It is suggested that the carbonyl group in parent xanthone 2 be reduced to hydroxyl group to investigate the complete structure-activity relationships. What is the mass and yield of each compound? Although it has been reported in the literature, a synthetic roadmap should also be added. Moreover, it is very strange for the authors to synthesize all the reported and known derivatives of the antifouling activities? The authors should explain in the manuscript.

Response 2: We thank the reviewer for suggesting to investigate the relevance of the carbonyl group at C-9. The aim of this work was an optimization of 3,4-dioxygenated xanthones, particularly at C-1; other positions were out of the scope of this SAR.

9-Hydroxy-xanthenes (xanthydrol derivatives) are not common metabolites of xanthones (most often the carbonyl group at C-9 xanthone derivatives is very stable; in contrast, the hydroxy group at C-9 furnishes very unstable derivatives that tend to disproportion into xanthenes and xanthones). For that reason, we did not consider this scaffold (without a carbonyl group at C-9) as promising; moreover, we have no information of synthetic xanthenes or xanthydrol itself with antifouling profiles, in contrast to several secondary metabolites of xanthones that have relevant antifouling activity and were the natural models that inspired this work of mimetics.  We added a synthetic roadmap for the synthesis since many readers are not acquainted with these derivatives. Most of the derivatives were available as a library of compounds in our lab; only the series of aminated xanthones were resynthesized. We have clarified this aspect not only in results and discussion but also in material and methods, with the corresponding yields added.

Point 3: According to the insights on the antifouling mode of action of the hit xanthones 21 and 23,it is recommended that the authors supplement molecular docking data for compounds and molecular targets associated with mussel adhesion, rather than merely conjectural.

Response 3: Such docking studies are currently very difficult to implement since we do not have enough information about the structure of these mussel adhesion proteins. Apart from the very specialized adhesion proteins, which can only be found in these organisms and therefore there are no homologous from other species available, for some not related adhesion proteins, or more ubiquitous proteins, we could use the information of homologous proteins, from other species to perform some docking studies. Nevertheless, this would be also highly speculative work since homologous proteins will differ greatly in terms of sequence and structure.

Point 4: "Synthesis" is not appropriate as a keyword. There is hardly any description of synthesis in the article. Please replace or supplement the part of synthetic chemistry.

Response 4: We agree. “Synthesis” was replaced by “anti-settlement” as one of the keywords.

Point 5: In the anti-settlement activity test, why choose such a high concentration as 50 μM for screening? It should decrease the concentration for screening, sometimes high concentration will inhibit the situation.

Response 5: Thanks for the comment. Our group has a vast experience in these bioassays with mussel larvae, and in our experience the acute response needs such high concentration to have a proper discrimination on what we may consider an effect on adhesion and/ or a response related to general toxicity. The bioassay is properly validated in our lab and widely published.

Point 6: Please add the MIC values of 1, 2, 4, 6, 8, 16, 19, 21, and obviously, different compounds exhibit different activity on different indicators, so if possible, all compounds should be tested for antimicrobial activity.

Response 6: MIC values were not possible to reach for the concentrations tested. To meet the reviewer concern we have moderated our interpretations on antibacterial activity and add information on dose-response bioassay on the revised manuscript.

It is true that the best approach is always to test all against everything, but in this case we had to make choices and we decided to test only our most promising compounds, selected from the macrofouling bioassay, to try to have a compound with complementary AF activity towards different levels of biofouling community.

Point 7: The “Materials and Methods” part is lack of “Antibacterial Activity”.

Response 7: Thanks for the identification of this lapse. The section “Antibacterial Activity” was added to Materials and Methods.

Point 8: There are some other problems that the authors should check carefully. Line 31/150: What does 15 refer to in the passage?  Please explain clearly. If it is different from the xanthones 15, it is suggested to use another expression method.

Response 8: Compounds numbers are always in bold. The value 15 is the recommended value for the LC50/EC50 ratio. This is a parameter set by the US Navy (see reference number 20 in the manuscript).

Point 9: Line 32: When the abbreviation first appears in the manuscript, please give the full name, for example, AF.

Response 9: The full name for AF - antifouling - was added to its first appearance, i.e, at line 86.

Point 10: Line 87/91: Please use the same expression, numbers or words. And check the number of xanthones.

Response 10: Sentence was changed to “Based on previous findings and structure-activity relationship (SAR) studies, in which we found that both hydroxy or methoxy groups at 3 and 4 position favour antifouling (AF) activity of xanthones, a library of 24 synthetic xanthones with three substitution patterns - 19 1,3,4-trisubstituted xanthones (1, 6, 8-10, 12-24), five 1,3,4,6-tetrasubstituted xanthones (2, 4, 5, 7, and 11) and one 1,2,3,4,6-pentasubstituted xanthone (3). Additionally, the chosen xanthones display a wide diversity of substituents at C-1 – three xanthones with a methyl group (1, 2, 3), four with halogenated alkyl groups (4-7), one with an hydroxymethyl group (8), three with an aldehyde (9-11), and 14 with diverse aminated alkyl groups (12-24) was selected and the derivatives 1-24 investigated for their antifouling potential (Figure 2).”.

The total number of xanthones was corrected to 24.

Point 11: Line 95: Why you choose these five kinds of bacteria for antibacterial assays? Please supplement the evidence.

Response11: Bacterial species were selected regarding their ability to produce biofilm, and among them, the ones that are more prominent in the biofouling film present here in the Portuguese coast, according to NGS studies from our group (Antunes et al., 2020). The 5 strains were also selected to be phylogenetically different, and to have a broader representation of different susceptibilities regarding AF promising compounds. This rational was included in the M&M section.

Point 12: Line 113: The results in Figure 3 are difficult to be compared between groups. An intuitive statistical method is recommended to compare differences between groups.

Response 12: The results are just compared with the negative control (Anova+Dunnet), not between groups. We understood the reviewer’ misinterpretation as the caption was confusing. We have rephrased the caption in the revised manuscript.

Point 13: Line 130: The difference of data between group C and group 16 cannot be shown in Figure 3. It is suggested to do it again and indicate the repetition times under the figure. The error lines of some groups are too high. Are the results reliable?

Response 13: As stated in the point 12, statistic comparisons were made between negative control and all the other conditions including positive control (C). Data from group C (positive control) and compound 16 are both 0% of settlement, meaning that both compounds inhibited 100% the larval settlement. The relatively high error is due to the 20 replicates used per condition, used to turn the assay most reliable.

Point 14: Line 172: It is recommended to mark significant differences of 6 and 16 in Figure 1. Why not set 12.5μL concentration in the concentration-response of antibacterial activity?  Supplements are suggested to validate the results.

Response 14: We have reformulated our interpretation and conclusions of the antibacterial bioassays to meet the reviewer concerns that we totally agree.

Point 15: Line 222: The names of proteins are not fully displayed, so the heat map of Figure 6A is suggested to be placed separately. Important proteins are recommended to be compared on separate heat maps.

Response 15: The heatmap from Figure 6A is just to show the whole picture regarding differentially expressed proteins that are represented by the color map. The most important (specific) proteins are, as suggested, represented separately in the Heatmap of Figure 8.

Point 16: Full text: It is recommended to mention figure or graph is analyzed in the result statement.

Response 16: Figures and graphs were cited throughout the entire manuscript.

Reviewer 3 Report

The manuscript is suitable to publish in this Journal, it should be accepted after minor revision.

  1. In the abstract and text, there are micro M, the micro g/mL are not necessary, please delete them, and table 1.
  2. There are many “Error! Reference source not found’, please revise them.

Author Response

The manuscript is suitable to publish in this Journal, it should be accepted after minor revision.

Point 1: In the abstract and text, there are micro M, the micro g/mL are not necessary, please delete them, and table 1.

Response 1: We thank for this suggestion. The values of EC50 are represented in Table 1 (and main text) both in μM and μg.mL-1 for comparison reasons. The μg.mL-1 value is needed to compare the obtained EC50 values with the US navy recommendation (EC50 < 25 µg.mL−1) . On the other hand, the EC50 values should also be represented in µM since molar concentrations gives a more accurate info about the compounds potency in relation to the molecular size and complexity.

Point 2: There are many “Error! Reference source not found’, please revise them.

Response 2: We were made aware by the Editor that these errors are editor's layout mistakes. They were already fixed.

Reviewer 4 Report

In this manuscript, Resende et al. reported the optimization of bioinspired xanthone derivatives with C-1 amination and 3,4-dioxygenation as efficient antifouling agents. Based on previous publication (Biomolecules 2020), a small library containing 24 xanthone derivatives has been prepared by organic synthesis, under the structural optimization from the original 3,4-dihydroxyxanthone together with chlorination, bromination and amination. The anti-fouling activity was evaluated using established inhibition assay of settlement of mussel larvae M. galloprovincialis. The first antifouling (AF) screening test and effectiveness test from xanthone library confirmed that the presence of a 3,4-dimethoxy substitution in the xanthone core and highlighted the importance of additional cyclic amine chain at C-1. Nine most promising AF xanthones didn’t show the significant inhibition against marine biofilm-forming bacteria, and most of them showed non-ecotoxic effect against marine crustacean Artemia salina. Then the potential antifouling mechanism of action of best two hits were evaluated by comparative proteomics, indicating the influence on adhesion mechanism of mussels. Interestingly, the down-regulated proteins from most effective xanthones from current research (myosin isoforms from pedal retractor muscle) are different from the down-regulated proteins from pyrano modification from previous research (proximal thread matrix protein 1b). Overall, this research presented the successful optimization of antifouling xanthones based on former SAR studies. The best hit from current research represent a high potential in future development for commercial antifouling agent. The identified potential protein target can be evaluated and might be useful in future to rationally design the inhibitors.

This manuscript represents an effective expansion based on former publication on the established bioassay and synthetic methodology. The hypothesis is very well developed from published data, and the experiments were designed properly with sufficient controls and replications. The data were analyzed under statistic evaluation. Finally, the conclusion is established based on proper discussion and match to the experimental results. Therefore, this manuscript is suitable to published on Marine Drugs.

However, that are still minors need to be improved, and list below for your attention:

1) Please go through the manuscript and correct the number of 25 to 24 when the library size of xanthones in this study was described.

2) Page 1, line 29/30: value EC50 = 7.28 and 3.57 µM; 3.03 and 1.26 µg/mL is reductant, please only keep µM value.

3) Page 7, line 204: ‘in expression of 160 and 159 proteins respectively…’, should be 156 and 159 proteins respectively…

4) Please clarify the antimicrobial activity: in abstract page 1, line 32/33: xanthones 6 and 16 were active against Roseobacter litoralis at 12.5 µM, but in text page 6 line 174, the interpretation is that they are not considered as antibacterial due to the lack of dose-dependence.

5) Would be possible to explain why the LD50 values is lower than the xanthones reported from Biomolecules 2020?

6) Would be possible to discuss the comparison between the best hit from current research with the best hit from Biomolecules 2020?

Thank you very much for your kind patience, hope the suggestions are helpful.

Author Response

However, that are still minors need to be improved, and list below for your attention:

Point 1:  Please go through the manuscript and correct the number of 25 to 24 when the library size of xanthones in this study was described.

Response 1: “xanthones 1-25” was changed to “xanthones 1-24” throughout the entire manuscript.

Point 2: Page 1, line 29/30: value EC50 = 7.28 and 3.57 µM; 3.03 and 1.26 µg/mL is reductant, please only keep µM value.

Response 2: We thank reviewer 4 for the suggestion. The values of EC50 are represented in Table 1 (and main text) both in μM and μg.mL-1 for comparison reasons. The μg.mL-1 value is needed to compare the obtained EC50 values with the US navy recommendation (EC50 < 25 µg.mL−1). On the other hand, the EC50 values should also be represented in µM since molar concentrations gives a more accurate info about the compounds potency in relation to the molecular size and complexity.

Point 3: Page 7, line 204: ‘in expression of 160 and 159 proteins respectively…’, should be 156 and 159 proteins respectively…

Response 3: ‘in expression of 160 and 159 proteins respectively…’, was changed to ‘in expression of 156 and 159 proteins respectively…’

Point 4: Please clarify the antimicrobial activity: in abstract page 1, line 32/33: xanthones 6 and 16 were active against Roseobacter litoralis at 12.5 µM, but in text page 6 line 174, the interpretation is that they are not considered as antibacterial due to the lack of dose-dependence.

Response 4: Although xanthones 6 and 16 presented a significant growth inhibition of the strain R. litoralis in the screening bioassays at 12.5 µM, an increase in bioactivity was not found on concentration-response study for higher concentrations of these compounds. Hence, these compounds were not considered as potent antibacterial agents, at least against these biofilm-forming strains. To meet the reviewer concern we have moderated our conclusions on antibacterial activity in the abstract and results sections.

Point 5: Would be possible to explain why the LD50 values is lower than the xanthones reported from Biomolecules 2020?

Response 5: The higher concentration tested in this study was 100 µM given the impossibility of producing some compounds in so high amount (for 500 µM test solutions) as in our previous study.

Point 6: Would be possible to discuss the comparison between the best hit from current research with the best hit from Biomolecules 2020? Thank you very much for your kind patience, hope the suggestions are helpful.

Response 6: This discussion was included in conclusions: “The substitution pattern of xanthones 21 and 23 induced higher anti-settlement activity against Mytilus larvae than our previously studied most promising and non-toxic synthetic xanthone, 3,4-dihydroxyxanthone, without losing the low toxicity to the environment, highlighting the benefits of the rational followed in this work.”

Round 2

Reviewer 2 Report

The revised manuscript could be accepted.

Author Response

The authors thank the valuable contribution of reviewer 2